# Using IVIM Parameters to Differentiate Prostate Cancer and Contralateral Normal Tissue through Fusion of MRI Images with Whole-Mount Pathology Specimen Images by Control Point Registration Method

**DOI:** 10.3390/diagnostics11122340

**Published:** 2021-12-12

**Authors:** Cheng-Chun Lee, Kuang-Hsi Chang, Feng-Mao Chiu, Yen-Chuan Ou, Jen-I. Hwang, Kuan-Chun Hsueh, Hueng-Chuen Fan

**Affiliations:** 1Division of Diagnostic Radiology, Department of Medical Imaging, Tungs’ Taichung Metroharbor Hospital, Taichung 43503, Taiwan; guojeng@gmail.com (C.-C.L.); t11925@ms.sltung.com.tw (J.-I.H.); 2Department of Medical Research, Tungs’ Taichung Metroharbor Hospital, Taichung 43503, Taiwan; kuanghsichang@gmail.com; 3Center for General Education, China Medical University, Taichung 404, Taiwan; 4General Education Center, Jen-Teh Junior College of Medicine, Nursing and Management, Miaoli 356, Taiwan; 5Department of Biomedical Engineering, National Yang Ming Chiao Tung University, Taipei 112, Taiwan; layzack121.be06@nycu.edu.tw; 6Division of Urology, Department of Surgery, Tungs’ Taichung Metroharbor Hospital, Taichung 43503, Taiwan; ycou228@gmail.com; 7Department of Radiology, National Defense Medical Center, Taipei 11490, Taiwan; 8Division of General Surgery, Department of Surgery, Tungs’ Taichung Metroharbor Hospital, Taichung 43503, Taiwan; sandersyeh@yahoo.com.tw; 9Department of Pediatrics, Tungs’ Taichung Metroharbor Hospital, Taichung 43503, Taiwan; 10Department of Life Sciences, National Chung Hsing University, Taichung 40227, Taiwan; 11Department of Rehabilitation, Jen-Teh Junior College of Medicine, Nursing and Management, Miaoli 356, Taiwan

**Keywords:** prostate cancer, Gleason score, intravoxel incoherent motion, whole-mount pathology, control point registration

## Abstract

The intravoxel incoherent motion (IVIM) model may enhance the clinical value of multiparametric magnetic resonance imaging (mpMRI) in the detection of prostate cancer (PCa). However, while past IVIM modeling studies have shown promise, they have also reported inconsistent results and limitations, underscoring the need to further enhance the accuracy of IVIM modeling for PCa detection. Therefore, this study utilized the control point registration toolbox function in MATLAB to fuse T2-weighted imaging (T2WI) and diffusion-weighted imaging (DWI) MRI images with whole-mount pathology specimen images in order to eliminate potential bias in IVIM calculations. Sixteen PCa patients underwent prostate MRI scans before undergoing radical prostatectomies. The image fusion method was then applied in calculating the patients’ IVIM parameters. Furthermore, MRI scans were also performed on 22 healthy young volunteers in order to evaluate the changes in IVIM parameters with aging. Among the full study cohort, the f parameter was significantly increased with age, while the D* parameter was significantly decreased. Among the PCa patients, the D and ADC parameters could differentiate PCa tissue from contralateral normal tissue, while the f and D* parameters could not. The presented image fusion method also provided improved precision when comparing regions of interest side by side. However, further studies with more standardized methods are needed to further clarify the benefits of the presented approach and the different IVIM parameters in PCa characterization.

## 1. Introduction

Prostate cancer (PCa) is the second most commonly diagnosed cancer and fifth most frequent cause of death among men worldwide [1,2]. The use of magnetic resonance imaging (MRI) for the clinical assessment of the prostate has been ongoing since the 1980s [3,4]. Relatedly, T2-weighted imaging (T2WI), diffusion-weighted imaging (DWI), and dynamic contrast enhanced (DCE) imaging have been combined to achieve multiparametric magnetic resonance imaging (mpMRI), and various studies have shown that mpMRI is the most accurate imaging approach in terms of both PCa detection and characterization [5,6,7].

In 2012, the Prostate Imaging Reporting and Data System (PI-RADS) guidelines were established by the European Society of Urogenital Radiology (ESUR) in order to standardize the acquisition, scoring, and reporting of prostate mpMRI, with PI-RADS version 2.1 being published in 2019 [8]. Since its establishment, a number of studies have shown the PI-RADS reporting system to have good diagnostic accuracy [9,10,11]. However, the PI-RADS system continues to show room for improvement. For example, a 2020 study by Woznicki et al. demonstrated one way in which the PI-RADS system might be improved upon, finding that an ensemble model combining a radiomics model with PI-RADS, prostate specific antigen density (PSAD), and digital rectal examination (DRE) models was superior to the PI-RADS model alone in terms of both PCa detection and clinical significance prediction [12].

In fact, a variety of models have been conceived and utilized over the years in order to enhance the efficacy of MRI techniques, with the intravoxel incoherent motion (IVIM) model originated by Le Bihan et al. being a prominent example [13]. In brief, the IVIM model has been applied to DWI, in order to better account for the effects of perfusion on diffusion [14]. More specifically, while DWI alone allows for the measurement of water molecule diffusion in biological tissues, the apparent diffusion coefficient (ADC) values yielded by DWI alone rely on a mono-exponential decay model that may mix extravascular molecular diffusion with the microcirculation of blood within capillaries (i.e., perfusion) in a manner that results in the flawed characterization of diffusion signals in biological tissues [15,16]. In contrast, the IVIM DWI model utilizes a biexponential decay function that allows for the separation and measurement of both molecular diffusion and capillary perfusion, which in turn provides further parameters; that is, in addition to ADC values–that can be used for tissue characterization, including the perfusion fraction (f), molecular diffusion coefficient (D), and perfusion-related diffusion coefficient (D*) [13,16,17]. Mathematically, the overall MRI signal attenuation, S/So, can be expressed in terms of IVIM parameters as follows [18]: S/S0 = f. exp [−b (D* + D)] + (1 − f) exp (−b.D)(1)
where S is the signal intensity, S0 represents the reference signal intensity at b = 0, f is the perfusion fraction, D is the molecular diffusion coefficient, and D* is the perfusion-related diffusion coefficient, and b is a value derived from the gradients used to acquired DWI scans 

In recent years, a number of studies have demonstrated the potential value of utilizing the IVIM model in the imaging of PCa. For example, various studies have found that the IVIM parameter D exhibits better performance than other parameters in differentiating the Gleason score (GS) of PCa lesions [19,20,21], while other studies have variously shown that both the D and ADC parameters are correlated with cell density in PCa and that valuable information regarding treatment response can potentially be gleaned from the changes in IVIM parameters over the course of treatment [22,23,24]. At the same time, a recent review by Brancato et al. observed that different studies have found inconsistent results for different IVIM parameters, noting, for example, that while several studies have found D* values to be significantly higher in tumor tissue than normal tissue, several others have found that D* values show no significant correlations [25]. 

One possible explanation of these inconsistent results is the imprecise registration of the PCa tissue locations on pathology slides with MRI images, which would result, in turn, in bias in the calculation of IVIM parameters. Several approaches are used to register of the histological PCa lesions to MR images, including registration the 3D histology volume with the 3D MR volume, registration of each 2D histology slice to its corresponding 2D MRI slice either by mutual information or by control points [26,27], or using 3D-printed of patient-specific-molds made from T2W-MRI images for prostate whole-mount sectioning [28]. In the present study, we utilized the control point registration toolbox function in the MATLAB in order to fuse the MRI images of PCa patients with corresponding whole-mount pathology slice images after carefully correlating the MRI images and the tumor margins delineated by a pathologist, as well as the prostate margins on both the MRI images and pathology specimen images, with the goal of this fusion being to eliminate potential bias in IVIM calculations stemming from the potential exclusion of tumor portions with no obvious signal changes. It was hoped that this would allow, in turn, for enhanced differentiation of PCa tissue with contralateral normal tissue. In addition, we also calculated the prostate IVIM parameters for a group of healthy young volunteers without PCa in order to compare the IVIM parameters of those healthy volunteers with those of the PCa patients.

## 2. Materials and Methods

This study was approved by the institutional review board (IRB) of our hospital. From October 2018 through December 2019, a total of 166 prostate patients underwent 3-Tesla (3 T) MRI in our department. Of those patients, 42 were confirmed to have PCa by either biopsy or surgery, and of those 42 patients, 25 who underwent biopsy before MRI were excluded from the study because the intraglandular bleeding resulting from biopsy could have led to bias. Moreover, 1 patient with a history of neoadjuvant hormone therapy was also excluded; ultimately, a total of 16 patients without biopsy prior to MRI were enrolled in the study. All of these patients received gadolinium contrast injection as dynamic enhancement, except for one who did not receive a gadolinium contrast injection due to impaired renal function. A total of 22 healthy young volunteers were also enrolled in the study as a control group. The selection criteria for the healthy young volunteers were no urinary tract disease, no previous history of prostate biopsy or operation, and no abnormal imaging finding on T2WI and DWI images. 

All of the MRI scans were performed on a Philips 3T Achieva scanner (Best, The Netherlands) using a 16-channel SENSE-XL-TORSO coil. The following parameters were used in T2W-MRI: TR: 4000–6000 ms, TE: 100 ms, thickness: 3 mm, acquisition voxel size: 0.7 × 0.4 mm^2^, number of signal averages (NSA): 3, and field of view (FOV): 160 × 160 mm. The following parameters were used in DW-MRI: TR: 2000–4000 ms, TE: 70 ms, thickness: 3 mm, acquisition matrix: 72 × 60 mm, NSA: 1, and FOV: 220 × 220 mm. Furthermore, a total of 10 different b values (0, 50, 100, 200, 400, 600, 1000, 1200, 1800, and 2000 s/mm^2^) were acquired. The MRI images of each participating patient were fused with the patient’s whole-mount pathology slice images using custom codes, which were implemented in MATLAB (R2017b, Mathworks, Natick, MA, USA), and the areas of PCa focus annotated by the pathologist in the pathology specimens were replicated on the corresponding MRI images (Figure 1), with the time interval between the MRI scan and the radical prostatectomy being less than 2 months for all the patients. 

To accurately register the regions of interest (ROIs) of PCa in the pathology slices to DW images, we used two steps. The first step consisted of the registration of pathology slices with T2WI images using the control point registration method (one of the toolbox function of MATLAB), Initially, the pathology slice containing the largest cancer tissue area and the corresponding MRI scan slice were chosen by experienced experts. Several control points were then selected on both T2WI and the histology image, with analogous points manually placed along the boundaries of the prostate and on identifiable landmarks (such as cavities in the transitional zone, urethra, or benign hyperplasia nodule) within the prostate. Between 5 and 8 control points were placed on the pathology slices and T2WI. The pathology slice image was then fitted with the T2WI image with these paired control points, and the non-linear transformation (fitgeotrans function of MATLAB) was used for the registration. In the second step, the ROI of the PCa was carefully replicated on the T2WI image according to the annotations made by pathologist on the histology slices. Due to the inconsistence between T2W and DW images, the mask of the ROI from T2WI was down sampled to the resolution of DW images, and thus the matrix alignment was conducted. Then, the ROI was easily placed onto DW images by overlapping them with the fused pathology/T2WI images. Using this control point registration approach, the image fusion process was performed with direct references provided by experienced experts, and thus potential error due to the automatic registration was eliminated, with the utilized method providing a clear visualization of the localization of PCa on DW images. The contralateral normal tissue was circled according to the PCa tissue location. More specifically, if the PCa tissue was located in the peripheral zone, then the normal tissue ROI was chosen from the peripheral zone on the opposite side, and if the PCa tissue was located in the transitional zone, then the normal tissue ROI was chosen from the opposing transitional zone tissue, with any areas of cystic degeneration in the transitional zone being carefully avoided after comparison with the pathology specimen. Furthermore, each lesion was characterized by its GS, and its malignant degree was defined as either high grade (GS > 4 + 3, N = 6) or low grade (GS < 3 + 4, N = 10). The cancer enhancement was determined by interpreter by comparing the pre- and post-enhanced images as well as dynamic contrast enhancement maps. IVIM parameters were acquired by the in-house software program implemented in MATLAB (R2017b, Mathworks, Natick, MA, USA) using the Bayesian shrinkage prior (BSP) fitting method. All statistical analyses were completed using SPSS version 25 (Corporation, Armonk, New York, NY, USA). IVIM data were calculated as a measure of the central tendencies of the IVIM values of PCa tissue and contralateral normal tissue. Significant differences in IVIM distributions between the two age groups were evaluated by applying the Student’s *t*-test and Mann–Whitney test, with the level of significance set at *p* < 0.05. 

## 3. Results

A total of 16 PCa patients were enrolled in the study. The mean age of those 16 patients was 68.1 years (range: 50–84 years), and their mean prostate-specific antigen (PSA) level was 9.62 ng/dL (range: 3.38–22.68 ng/dL; SD: 4.98). In terms of GS, 10 of the patients had a GS of 7, 3 of the patients had a GS of 8, and 3 of the patients had a GS of 9. In addition, 22 healthy young volunteers were also enrolled in the study as a control group, and their mean age was 35.64 years (range: 24–49 years).

To determine which IVIM parameters of the transitional zone and peripheral zone, if any, exhibited significant differences due to aging in the healthy volunteers, the healthy volunteers were divided into two age groups (those younger than 35 years and those older than 35 years), and the data for the two age groups were compared using the two-independent-samples test and the Mann–Whitney test. The results showed that only the f and D* parameters in the transitional zone and the D* parameter in the peripheral zone were significantly correlated with age (Table 1). 

Furthermore, the data for all the 22 healthy volunteers were analyzed using the paired samples test in order to determine which parameters, if any, differed significantly between the transitional zone and the peripheral zone, and the results indicated that only the ADC parameter differed significantly between the two zones (Table 2)

To determine which IVIM parameters of the transitional zone and peripheral zone, if any, exhibited significant differences due to aging among all the study participants, that is, both the 22 healthy volunteers and the 16 PCa patients (*n* = 38, age range: 24–84 years), the data for the full study cohort were compared using the Spearman correlation coefficient. The results showed that the f values in both the transitional and peripheral zones were significantly increased as age increased and that the D* values in both the transitional and peripheral zones were significantly decreased as age increased (Table 3).

In addition, to determine various measures of diagnostic accuracy in the differentiation of PCa tissue and normal tissue–namely, the cut-off value, area under curve (AUC), sensitivity, specificity, positive predictive value (PPV), negative predictive value (NPV), accuracy, and age-adjusted odds ratios (ORs) for the cut-off values of each of the IVIM parameters–the data for all the study participants were analyzed using the Mann–Whitney test (Table 4). The results showed ADC was the most accurate parameter in differentiating PCa tissue and normal tissue, with 87.5% sensitivity, 93.8% specificity, and 90.62% accuracy.

The 16 PCa patients were divided into two groups according to GS, with the 10 patients with a GS of 7 in one group, and the 6 patients with a GS of 8 or 9 in the other group. The data for the two groups were then compared using the two-independent-samples test and the Mann–Whitney test to determine which IVIM parameters of PCa lesions, if any, were significantly correlated with GS. The results indicated that the D* parameter was significantly correlated with GS. Furthermore, the 15 patients who received gadolinium injection for dynamic enhancement were divided into two groups depending on whether or not their PCa lesion exhibited enhancement, with 11 patients exhibiting PCa lesion enhancement and 4 exhibiting no enhancement. The data for the two groups were then compared using the two-independent-samples test and the Mann–Whitney test to determine which IVIM parameters of PCa lesions, if any, were correlated with enhancement. The results indicated that none of the parameters were significantly correlated with enhancement. Lastly, the IVIM parameter data for the PCa tissue and the contralateral normal tissue of the 16 PCa patients were compared using the paired *t* test in order to determine which of the parameters, if any, could differentiate the PCa tissue from the normal tissue. The results indicated that the D and ADC parameters differed significantly between the PCa tissue and the contralateral normal tissue, while the f and D* parameters exhibited no significant difference between the two types of tissue (Table 5).

## 4. Discussion

A number of models have been proposed and utilized for the analysis of MRI data used for PCa screening, and various studies have been conducted to determine the relative accuracy and efficacy of these models. For example, Liu et al. recently conducted a study comparing the potential of four models, namely, a mono-exponential model, kurtosis model, IVIM model, and IVIM–kurtosis model, in diagnosing and assessing the aggressiveness of PCa, with their results suggesting similar diagnostic efficacy levels for all four models but possible superiority of the IVIM–kurtosis model for assessing tumor aggressiveness [29].

At the same time, a number of technical factors may also impact the accuracy of MRI models. For example, a recent study by Malagi et al. found that their novel model combining a total variation (TV) penalty function with the traditional IVIM-diffusion kurtosis imaging (DKI) model at 1.5 T showed lower estimation errors in the clinical characterization of PCa and benign prostatic hyperplasia than the traditional IVIM-DKI model at 3 T. In other words, varying the technical factor of the magnetic field strength from 3 T to 1.5 T while also applying the TV penalty significantly enhanced the accuracy of the traditional IVIM-DKI model, suggests that the resulting novel model could potentially be utilized for improved prostate lesion detection [30]. Meanwhile, other studies have shown that still other factors, including the number and combination of b values used and the echo time used, can also affect the accuracy of IVIM models [31,32]. In short, previous studies suggest a number of means through which the analysis of MRI data for PCA detection and differentiation could potentially be further refined and improved upon, including the development and application of new models and combinations of model, as well as adjustments to technical factors.

With such issues in mind, we reviewed a large number of whole-mount pathology specimens and corresponding MRI images, and found that there were usually some discrepancies between the signal change regions on the MRI images and the true PCa lesions. More specifically, it is not uncommon for portions of PCa lesions to exhibit no obvious signal changes on MRI images, meaning that if tumor margins are determined based solely on signal changes, those portions of a tumor showing no signal change would not be included in the calculation of IVIM parameters, resulting in bias. Therefore, we utilized the control point registration method in the present study, which entailed using custom codes implemented in MATLAB to fuse MRI images with corresponding whole-mount pathology slice images through affine transformation, in order to decrease such potential bias in IVIM parameters. Relatedly, the rationale for fusing the given histology specimen image with the T2WI image first and then placing the ROI on the corresponding DWI image was that the margin of the DWI image was usually distorted and low resolution, making it difficult to compare the margin of the DWI image with that of the pathology specimen image. As such, it was expected that the presented image fusion method would yield more accurate IVIM parameters, which would in turn potentially allow for enhanced differentiation of PCa tissue from contralateral normal tissue. 

Although there are several registration methods mentioned above, which might help the laborious and time-consuming histology-MRI slice correspondence jobs performed by experienced radiologists and pathologists, these registration processes still need some manual adjustment during the registration process. These studies also utilize the ground truth slices determined by experienced experts as validation [27]. Customized 3D printed molds based on pre-operative MRI improves the accuracy of registration, but this approach requires a change in clinical protocol that is not practical in most hospitals. Therefore, the registration of 2D histology slices to corresponding 2D MRI images is more practical for our clinical routine. Manual selection of landmarks for registration is still the reliable and commonly adopted approach [27]. The manual control point registration method has also been used in previous studies [33,34]. Furthermore, study of IVIM parameters requires mapping the extent of PCa with DWI-MRI, because of low resolution of DWI images, direct registration of PCa area to DWI is difficult. We therefore imported T2WI and DWI images into MATLAB simultaneously, after fusion of pathology slices and T2WI images, overlapping them with corresponding DWI images provided an easy way to place the ROIs on the DWI images. 

One finding of the present study was that, among the full study cohort of both the 22 healthy volunteers and the 16 PCa patients, the f values in both the transitional and peripheral zones were significantly increased as age increased and the D* values in both the transitional and peripheral zones were significantly decreased as age increased. Meanwhile, we further found that among the 22 young and healthy volunteers, only the f values in the transitional zone and the D* values in both the peripheral and transitional zones were significantly different in the two age groups (age < 35 y/o and age > 35 y/o). According to Le Bihan’s hypothesis [13], the f value represents the blood flow in the microcapillary environment and is considered as a parameter of pseudo-diffusion, so the finding of the present study that the f values were increased along with aging process in present study may be reflecting the angiogenesis process (which results in increased perfusion) that occurs during prostatic hyperplasia as age increases. This finding is consistent with a 2017 study by Shi et al. [35]. However, the present study’s finding that D values not significantly increased or decreased during aging process, but D* values decreased as age increased, is somewhat inconsistent with the finding of Shi et al. reporting that D values were significantly increased with age, while ultrahigh ADC values were significantly decreased after 50 years of age. However, the D* values have shown poor reproducibility in previous studies [35,36]. In contrast with D* values, meanwhile, f values might constitute a more consistent and reliable parameter in terms of reflecting the angiogenesis process in prostate hyperplasia during aging. Otherwise, there were a number of important differences between the earlier Shi et al. study and the present one. For example, the Shi et al. study included only healthy participants with normal prostates, whereas the analyzed normal tissue regions of our participants more than 50 years old were chosen after examining their pathology specimens to avoid PCa tissue contamination. The Shi et al. study also had a larger number of participants overall (*n* = 67 as opposed to *n* = 38 for the present study) and considered scanned parameters with DWI utilizing 15 b values ranging from 0 to 3000 (as opposed 10 b values ranging from 0 to 2000), among other differences. As such, it is possible that the differing findings of the two studies may be due to their substantially differing methodologies and populations, such that further studies using consistent methodologies may be required to clarify the changes in prostate IVIM parameters with age and thereby further improve the early detection and diagnosis of PCa.

Another finding of the present study was that, among the participants with PCa, the D and ADC parameters could differentiate PCa tissue from contralateral normal tissue, while the f and D* parameters could not. These results seem only partially consistent with those of numerous past studies considered in a recent review by Brancato et al., which generally found significant differences between PCa tissue and normal tissue for f as well as D [25]. Moreover, while the present study’s finding that D* could not differentiate between normal and PCa tissues was consistent with those of a number of past studies (for example, those by Ueda et al. [37] and Feng et al. [38]), it was also inconsistent with still other past studies (such as those by Pesapane et al. [16], Kuru et al. [19], and Valerio et al. [39]) which found that D* values were significantly lower in normal tissue than in tumor tissue. Here again, however, because of the lack of a standardized protocol for DWI studies, including the number of selected b-values and the echo time used, as well as population and other methodological differences among studies, it seems impossible at present to draw firm conclusions as to which parameters can or cannot reliably differentiate PCa tissue from normal tissue.

Relatedly, the present study further found that only the D* parameter was significantly correlated with the GS values of the PCa patients, a finding which is inconsistent with a number of past studies (such as those by Zhang et al. [20], Kim et al. [21], and Barbieri et al. [40], among others) finding instead that D and ADC, but not D*, exhibit significant differences between low- and high-grade lesions. Again, however, methodological differences between those earlier studies and the present study make it difficult to draw firm conclusions as to which findings are accurate. For example, it may be that the novel image fusion methodology used in the present study provides more accurate IVIM parameter values by eliminating potential bias, such that its finding that only the D* parameter is significantly correlated with GS values is, in fact, accurate. On the other hand, in addition to having larger sample sizes than the present study, the Zhang et al., Kim et al., and Barbieri et al. studies all used various GS scores to group their participants and, in turn, define low- versus high-grade tumors, adding further uncertainty in comparing the findings of the those studies with one another and with those of the present study. 

Otherwise, it is also worth noting that some previous studies have used MRI fusion biopsy or non-targeted systemic biopsy results to determine the exact tumor locations of the included patients [39,41] or have included patients who underwent biopsy within the preceding few months [16]. The latter approach could interfere with IVIM calculations because a post-biopsy hematoma can last for up to several months and appear on T1WI images as patchy hyperintensity, while the former approach cannot precisely define the tumor location and could cause the location of the ROI to be inaccurate. As such, by avoiding those issues, the present study may have provided more accurate IVIM results.

Together, the results of the present and past studies, as well as the methodological differences among them, serve in large part to underscore the importance of standardizing DWI study protocols in order to improve their repeatability and reproducibility. This would allow for more fair and robust comparisons across studies that would, in turn, aid greatly in clarifying the benefits of mpMRI and specific IVIM parameters for PCa detection and characterization.

One limitation of the present study is that only a small number of PCa patients (16) were included. This is because the public health insurance system in Taiwan only provides coverage for MRI scans of the prostate after a patient has undergone a prostate biopsy. Moreover, in a patient who has undergone a biopsy before an MRI scan, the hematoma resulting from the biopsy would subsequently be retained within the prostate tissue for many months and would interfere with any MRI data. Another limitation was that we did not measure the PSA levels of our non-patient volunteers. So, while most of the volunteers were relatively young, we cannot prove that there was no possibility of occult PCa among the volunteers. However, there was no evidence of PCa in their imaging data. Another limitation was that it was difficult to perfectly match the MRI images with the pathology slice images. For example, although each axial MRI scan was performed, in accordance with our hospital’s protocol, perpendicular to the urethra, the angles of the whole-mount pathology specimen slices were not consistent in every case (out-of-plane alignment problem). 3D registration might be a solution to solve this limitation in the future work. Furthermore, during the preparation of the whole-mount pathology slices, the tissue loss and tissue distortion could have significantly altered the appearance of the original histological scene. In addition, some of the cancer foci were very small, which was difficult to address on the MRI images, especially on DWI images, which are acquired with low resolution matrix, and the partial volume effect would enhance this problem, and cause a poor evaluation on a small ROI. Therefore, we only chose the largest lesion in each case. Even if a manual control point registration was implemented to improve the lesion targeting, there is still a pitfall with our method. The spatial shift on the through-plane could not be corrected because of 2D registration. Furthermore, due to the downsampling of the ROI mask, it yields an infidelity problem of the contouring. This would produce the error of pixel selection within the ROI, especially around the boundary of the prostate. Usually, there is a geometrical difference among slicing of pathology and MR images. 

Nonetheless, the results of this study showed that the presented method utilizing a control point registration method to fuse MRI images with pathology slice images provided improved precision and reduced bias when comparing ROIs side by side. Moreover, the results indicated that the D and ADC parameters could differentiate PCa tissue from normal tissue, which has also been found in previous studies. Together, these results suggest that IVIM parameters and the presented image fusion method could potentially be introduced into the PI-RADS system as an additional diagnostic tool. Again, however, further studies with more standardized DWI protocols and methods should be undertaken in order to further clarify the benefits of the presented image fusion approach and the different IVIM parameters for PCa detection and characterization.

## Figures and Tables

**Figure 1 diagnostics-11-02340-f001:**
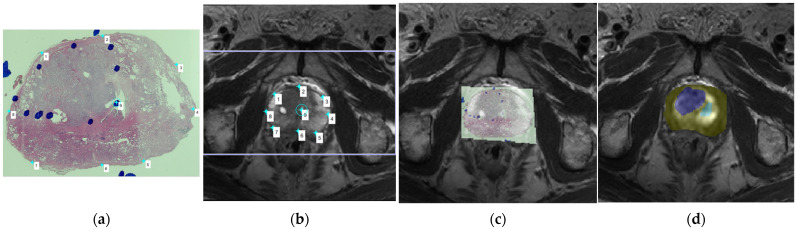
(**a**) A whole-mount prostate slice image. The locations of prostate cancer (PCa) tissue were first indicated as blue circles drawn by the pathologist; (**b**) We then compared the whole-mount slice image and the T2-weighted imaging (T2WI) images, choosing the corresponding T2WI image in order to determine the placement of corresponding anchors on the margin and internal landmarks in both the pathology slice image and the magnetic resonance imaging (MRI) image; (**c**) The pathology slice image and the MRI image were then fused together, with the locations of PCa tissue indicated by the blue circles overlapping with the corresponding areas of the MRI image; (**d**) The region of PCa tissue was then carefully drawn on the DWI image, which was at the same level as the selected T2WI image, as the blue region of interest (ROI), and the contralateral normal tissue was drawn as the green ROI.

**Table 1 diagnostics-11-02340-t001:** Comparison of age and IVIM parameters between the two groups of young volunteers.

Mean ± SD or Median (Q1,Q3)		Age < 35y/o (*n* = 11)	Age > 35y/o (*n* = 11)	*p*
Age (years)	Mean ± SD	29.18 ± 3.34	42.09 ± 4.76	<0.001
f (%)	TZ	34.7 (29.21, 41.95)	44.26 (38.50, 58.09)	0.02 *
	PZ	37.33(27.68, 40.58)	39.30 (35.86, 64.26)	0.101
D (×10^−3^ mm^2^/s)	TZ	0.84 (0.792, 0.94)	0.70 (0.63, 0.82)	0.07
	PZ	0.84 (0.80, 0.93)	0.78 (0.72, 0.88)	0.133
D* (×10^−3^ mm^2^/s)	TZ	5.41 (4.18, 8.21)	3.68 (3.04, 5.66)	0.016 *
	PZ	6.26 (5.51, 7.23)	4.14 (3.13, 5.99)	0.034 *
ADC (×10^−3^ mm^2^/s)	TZ	1.45 (1.26, 1.57)	1.45 (1.20, 1.55)	0.939
	PZ	1.46 (1.33, 1.71)	1.65 (1.45, 1.99)	0.193

Q1: 25th percentile. Q3: 75th percentile. TZ: transitional zone. PZ: peripheral zone. * indicates significant *p* value.

**Table 2 diagnostics-11-02340-t002:** Comparison of IVIM parameters between transitional zone and peripheral zone in volunteer group.

		f (%)	*p*	D (×10^−3^ mm^2^/s)	*p*	D* (×10^−3^ mm^2^/s)	*p*	ADC (×10^−3^ mm^2^/s)	*p*
Zone	TZ	38.8 (34.6, 44.8)	0.437	0.80 (0. 68, 0. 85)	0.198	4.41 (3.63, 5.95)	0.069	1.45 (1.25, 1.56)	0.016 *
	PZ	37.4 (34.1, 50.2)		0.82 (0.77, 0.89)		5.80 (3.93, 6.96)		1.52 (1.36, 1.74)	

TZ: transitional zone. PZ: peripheral zone. *p* for Mann–Whitney test. * indicates significant *p* value.

**Table 3 diagnostics-11-02340-t003:** Correlations between IVIM parameters and aging.

		Spearman Correlation Coefficient	*p*
f (%)	TZ	0.471	0.003 ^a^
	PZ	0.411	0.010 ^a^
D (×10^−3^ mm^2^/s)	TZ	−0.152	0.361
	PZ	−0.059	0.725
D* (×10^−3^ mm^2^/s)	TZ	−0.339	0.037 ^a^
	PZ	−0.436	0.006 ^b^
ADC (×10^−3^ mm^2^/s)	TZ	0.304	0.064
	PZ	0.272	0.099

TZ: transitional zone. PZ: peripheral zone. ^a^
*p* < 0.05. ^b^
*p* < 0.01. *p* for Spearman’s rank correlation coefficient.

**Table 4 diagnostics-11-02340-t004:** Measures of diagnostic accuracy for IVIM parameters.

	Cut-off Value	AUC (95% CI)	Sen(%)	Spe(%)	PPV(%)	NPV(%)	Accuracy	Age adj. OR	95% CI
f (%)	48.02	0.68 (0.49–0.92)	75	68.8	70.58	73.33	71	0.15	0.032–0.713
D (×10^−3^ mm^2^/s)	0.55	0.88 (0.75–1.00)	62.5	100	100	72	81	N/A	N/A
D* (×10^−3^ mm^2^/s)	3.08	0.60 (0.40–0.80)	31.3	100	100	59	65	N/A	N/A
ADC (×10^−3^ mm^2^/s)	1.43	0.89 (0.74–1.00)	87.5	93.8	93.3	88.2	90.62	0.10	0.001–0.117

AUC: area under curve. Sen: sensitivity. Spe: specificity. PPV: positive predictive value. NPV: negative predictive value. CI: confidence interval. Age adj. OR: age-adjusted odds ratio. N/A: not available.

**Table 5 diagnostics-11-02340-t005:** Comparison of IVIM parameters between PCa/normal tissue, two Gleason score groups (7 vs. 8 + 9), and non-enhanced vs. enhanced PCa tissue.

		f (%)	*p*	D (×10^−3^ mm^2^/s)	*p*	D* (×10^−3^ mm^2^/s)	*p*	ADC (×10^−3^ mm^2^/s)	*p*
PCa vs. normal	PCa	42.7 (35.6, 49.8)	0.086	0.51 (0.36, 0.60)	0.001 *	3.73 (2.96, 6.37)	0.439	1.03 (0.81, 1.37)	<0.001 *
	Normal	52.1 (41.9, 55.2)		0.80 (0.70, 1.07)		4.31 (3.34, 5.21)		1.75 (1.51, 2.04)	
Gleason score	≤7 (*n* = 10)	45.2 (38.5, 51.1)	0.338	0.48 (0.39, 0.55)	0.256	3.33 (2.92, 4.82)	<0.001 *	1.01 (0.82, 1.20)	0.118
	>7 (*n* = 6)	35.8 (33.8, 49.4)		0.56 (0.28, 0.93)		5.40 (3.36, 8.10)		1.33 (0.66, 2.27)	
Enhanced	No (*n* = 4)	49.8 (33.9, 58.4)	0.116	0.53 (0.43, 1.10)	0.224	3.17 (2.67, 9.76)	0.776	1.21 (1.08, 1.97)	0.341
	Yes (*n* = 11)	38.5 (35.3, 45.6)		0.51 (0.34, 0.62)		4.36 (3.27, 6.43)		1.00 (0.81, 1.41)	

*p* for Mann-Whitney test. * indicates significant *p* value.

## Data Availability

Not applicable.

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
