# Peer review of "Using IVIM Parameters to Differentiate Prostate Cancer and Contralateral Normal Tissue through Fusion of MRI Images with Whole-Mount Pathology Specimen Images by Control Point Registration Method"

_diagnostics, 2021, doi:10.3390/diagnostics11122340_

Round 1

Reviewer 1 Report

In their manuscript "Analysis of Aging Changes in IVIM Parameters and Differences between Prostate Cancer and Contralateral Normal Tissue through Fusion of MRI Images with Whole-mount Pathology Specimen Image", Lee et al. describe an approach for fusion of MRI-pathology images with the aim of analyzing aging changes and differences in PCa and contralateral prostatic tissue in IVIM parameters. The results are quite interesting and might contribute to the advancement of IVIM parameters in clinical practice. However, there are some major issues that should be addressed by the authors to make the manuscript potentially acceptable:

Abstract

lines 27-29: the sentence seems not to be complete. Please, complete the sentence for example as following: ...the accuracy of IVIM modeling for...

lines 32-33: please rephrase

The image fusion method should be better highlighted and described. Please, reorganize the abstract accordingly.

Introduction:

The introduction is too long, in particulary in the first part (line 45-76). The introduction on prostate cancer, PIRADS and MRI in general is too long. It should be more appropriate to focus on diffusion MRI and Non-gaussian diffusion models, of which IVIM model.

I think that the authors should better highlight the importance of pathology-MRI fusion, methods already developed and explore literature on this topic. Being the nature of the manuscript more methodologic, you should enforce this aspect.

It is not clear from the title what is the main aim of the manuscript. Is the first aim to analyze aging changes in IVIM parameters or to assess differences between PCa and normal tissue by means of a method of MRI-pathology fusion? The title and introduction should be reorganized accordingly.

Materials and Methods

The MRI-pathology fusion method should be better described. Did the authors used a coregistration method? Pòease, clarify.

In case of coregistration or alignment method, did the authors assess the quality of MRI-pathology registration method by means of quantitative measures?

I wonder if the clinical outcome of age is intended only for differentiating among healthy volunteer or differences could be assessed also among PCa patients.

Results

Review results based on materials and methods comments.

Discussion

Discussions should be abbreviated and revised according to previous comments. A more consistent discussion should be performed on methods for MRI-pathology fusion. Some references:

https://doi.org/10.1016/j.compmedimag.2010.12.003

10.7937/K9/TCIA.2016.TLPMR1AM

https://arxiv.org/abs/1907.00324 

Reviewer 2 Report

Intravoxel incoherent motion (IVIM) DWI is a relatively old concept described by Bihan in 1988 that has currently been reused in the screening for prostate cancer using more powerful 3T machines. I must congratulate the authors for their work. The methodology is solid with good statistics and results

No comments from my part.

Author Response

Point 1:  Intravoxel incoherent motion (IVIM) DWI is a relatively old concept described by Bihan in 1988 that has currently been reused in the screening for prostate cancer using more powerful 3T machines. I must congratulate the authors for their work. The methodology is solid with good statistics and resultsNo comments from my part.

Response 1:  Thank you. Your consideration and comments are greatly appreciated. 

Round 2

Reviewer 1 Report

All major comments have been addressed. I suggest further minor revision for english language and style.